# A structured RNA motif locks Argonaute2:miR-122 onto the 5′ end of the HCV genome

Luca F. R. Gebert [1], Mansun Law [2] & Ian J. MacRae [1✉]

microRNAs (miRNAs) form regulatory networks in metazoans. Viruses engage miRNA networks in numerous ways, with *Flaviviridae* members exploiting direct interactions of their RNA genomes with host miRNAs. For hepatitis C virus (HCV), binding of liver-abundant miR-122 stabilizes the viral RNA and regulates viral translation. Here, we investigate the structural basis for these activities, taking into consideration that miRNAs function in complex with Argonaute (Ago) proteins. The crystal structure of the Ago2:miR-122:HCV complex reveals a structured RNA motif that traps Ago2 on the viral RNA, masking its 5′ end from enzymatic attack. The trapped Ago2 can recruit host factor PCBP2, implicated in viral translation, while binding of a second Ago2:miR-122 competes with PCBP2, creating a potential molecular switch for translational control. Combined results reveal a viral RNA structure that modulates Ago2:miR-122 dynamics and repurposes host proteins to generate a functional analog of the mRNA cap-binding complex.

[1] Department of Integrative Structural and Computational Biology, The Scripps Research Institute, La Jolla, CA 92037, USA. [2] Department of Immunology and Microbiology, The Scripps Research Institute, La Jolla, CA 92037, USA. ✉email: macrae@scripps.edu

Mammalian Argonaute (Ago) proteins use ~22 nt short RNA guides to target partially complementary sites generally found in the 3' UTRs of mRNAs, recruiting additional factors that promote post-transcriptional gene silencing through translational repression and induction of mRNA decay[1]. miRNAs form a complex regulatory network, modulating diverse pathways in metazoan biology. Misregulation in miRNA activity is connected to developmental defects[1] and cancer[2].

While the animal miRNA machinery appears to derive from a primal antiviral RNA silencing mechanism[3], the relentless tides of viral evolution have carved out a multitude of ways in which host miRNA pathways are exploited to benefit viral fitness[4]. Pro-viral adjustment of the host expressome is achieved by viral-encoded miRNAs (e.g. KSHV[5]) or modulation of host miRNA activity (e.g. HPV[6]), including retargeting of host miRNA complexes to cellular mRNAs[7]. The most direct, yet poorly understood, mechanisms involve binding of host miRNAs to genomes of single-stranded RNA viruses of the *Flaviviridae* family: HCV (miR-122[8,9]), BVDV (miR-17[10]), and ZIKV (miR-21[11]) exploit host miRNAs to increase viral fitness through direct interactions with regions within their RNA genomes rich with secondary structure.

The interaction between the HCV genome and the liver-abundant miRNA miR-122 was the first direct viral RNA-host miRNA interaction to be discovered and is the best-studied to date[8,9]. HCV is a positive-strand single-stranded RNA virus that affects approximately 71 million people worldwide. In most patients the acute infection progresses to chronic liver infection, negatively affecting liver function and increasing the risk for developing hepatocellular carcinoma[12]. The virus' decade-long persistence in the host is maintained through a delicate balance of genome replication, translation, RNA decay, and maintenance of viral subcellular compartments[13].

Stabilization of HCV RNA was reported through two tandem miR-122 complementary sites at the very 5' end of the viral genome[8,9], and involves the host protein Argonaute2 (Ago2)[14] (Fig. 1a). Additionally, miR-122 binding modulates viral replication and polyprotein translation[15], and affects folding of the viral Internal Ribosomal Entry Site (IRES)[16,17]. The binding of miR-122 to the HCV genome leads to de-repression of host mRNAs normally targeted by miR-122, suggesting a causal link between chronic HCV infection and increased risk of developing liver cancer[18]. The diminutive 5'-terminal tail of the HCV genome thus acts as a regulatory epicentre, involved in viral RNA stability and folding, replication, translational control, and perturbation of host gene expression, that ultimately determines viral pathogenesis.

While the field agrees on the stabilizing effect of Ago2:miR-122 binding to the HCV 5'-terminus, which has been identified as a therapeutic target[19], the mechanisms promoting stabilization and the relationship between the two tandem sites, including their roles during the viral life-cycle, are poorly understood. Moreover, it is unclear how HCV RNA structures, not observed in characterized cellular miRNA-binding sites, might influence interactions with Ago2.

Here, we determine the structural basis for recruitment of miR-122 to the HCV 5'-terminus, and biochemically examine how recruitment shapes interactions with other host factors. The 3.0 Å resolution crystal structure of the Ago2:miR-122:HCV RNA complex reveals a structured motif in the viral RNA that exploits a previously unrecognized nucleotide-binding pocket within a conserved region of Ago2. The motif locks Ago2 in a widened conformation, which counteracts the propensity of Ago2 for lateral diffusion on target RNAs, leading to robust protection of the viral RNA 5' end from enzymatic attack. We also examined interactions between the HCV 5' terminus, Ago2, and host protein PCBP2, which was suggested to bind HCV RNA in competition with miR-122 and promote viral translation[15]. Our data indicate a more nuanced model in which one copy of Ago2:miR-122 remains anchored on the viral RNA 5' end while a second copy competes with PCBP2 for the adjacent miR-122 binding site. Altogether, these data reveal the molecular architecture of a structured miRNA recognition site, how Ago2:miR-122 binding protects the HCV genomic RNA, and a potential molecular switch for controlling the balance between viral protein translation and RNA replication.

## Results

**Ago2 occupancy at miR-122 Site-1 protects HCV RNA from XRN1.** The miR-122 binding region stabilizes the viral RNA[8,9] through protection from cellular exoribonucleases[14,20,21], primarily exoribonuclease 1 (XRN1)[22]. In vitro biochemistry corroborated a role for miR-122 alone[23], but it is widely recognized that Ago proteins change the binding properties of miRNAs[24–27]. We, therefore, tested whether the purified Ago2:miR-122 complex is able to protect the HCV 5' UTR from recombinant XRN1. In a control reaction, XRN1 rapidly degraded a purified RNA representing the HCV 5' UTR (Fig. 1b). However, pre-incubation with Ago2:miR-122 led to a ≥400-fold increase in the half-life of the viral RNA (Fig. 1b). Mutation of miR-122 Site-1 to prevent binding by Ago2:miR-122 (Supplementary Fig. 1a–c) abolished protection from XRN1 (Fig. 1c). Surprisingly, even though XRN1 is an exonuclease and thus attacks the 5' end of its RNA substrates, preventing Ago2:miR-122 binding at Site-2 also made the 5' UTR more susceptible to XRN1, albeit to a much smaller extent (~2.5-fold decrease in half-life) (Fig. 1c). Full protection from XRN1 could be restored by using Ago2 loaded with a mutant (MUT) form of miR-122 that recognizes the mutated sites (Supplementary Fig. 1d, e). We conclude Ago2 occupancy at Site-1 protects the HCV 5' UTR from XRN1 and this protection is augmented by Ago2 occupancy at Site-2.

**SL1 is required for full protection against XRN1.** Site-1 contains a stem-loop structure (stem-loop 1, SL1) that physically separates predicted base-paired regions between miR-122 and the HCV RNA (Supplementary Fig. 1b). From a miRNA-targeting perspective, SL1 is highly unusual as miRNA binding sites are typically devoid of secondary structure[28]. We wondered if this viral RNA structure contributes to protection from XRN1. We prepared a mutant 5' UTR RNA containing a 4-adenosyl linker in the place of SL1 (ΔSL1). Removal of SL1 led to an ~4-fold reduced half-life of the RNA in our nuclease assay compared WT (Fig. 1d). Surprisingly, combining the ΔSL1 mutation with a mutation that prevents Ago2 binding at Site-2 (MUT2) abolished protection from XRN1 (Fig. 1d), despite the observation that the ΔSL1 Site-1 is fully competent for Ago2-binding (Supplementary Fig. 1f).

To test if this unique RNA feature plays a similar role in infected cells we prepared genomic viral RNA, based on the H77 deltaE1p7 sequence[29], with the ΔSL1 mutation. As controls, we also produced wildtype (WT) viral RNA, and viral RNAs with mutations that disrupt miR-122 binding either Site-1 (MUT1) or Site-2 (MUT2). Viral RNAs were electroporated into Huh-7.5.1 cells and total RNA was isolated at time points spanning 5 days. Analysis by RT-PCR revealed all mutant RNA levels steadily declined, diminishing to ~1% of the WT RNA level after five days, demonstrating that Site-1, Site-2, and SL1 each contribute to the accumulation of viral RNA in cells (Fig. 1e).

The 5' terminus of HCV is required for replication[30], and we observed that a replication-deficient mutant, with catalytic residues in NS5B mutated from GDD > AAG[20], exhibited a decline in viral RNA similar to the SL1, MUT1, and MUT2

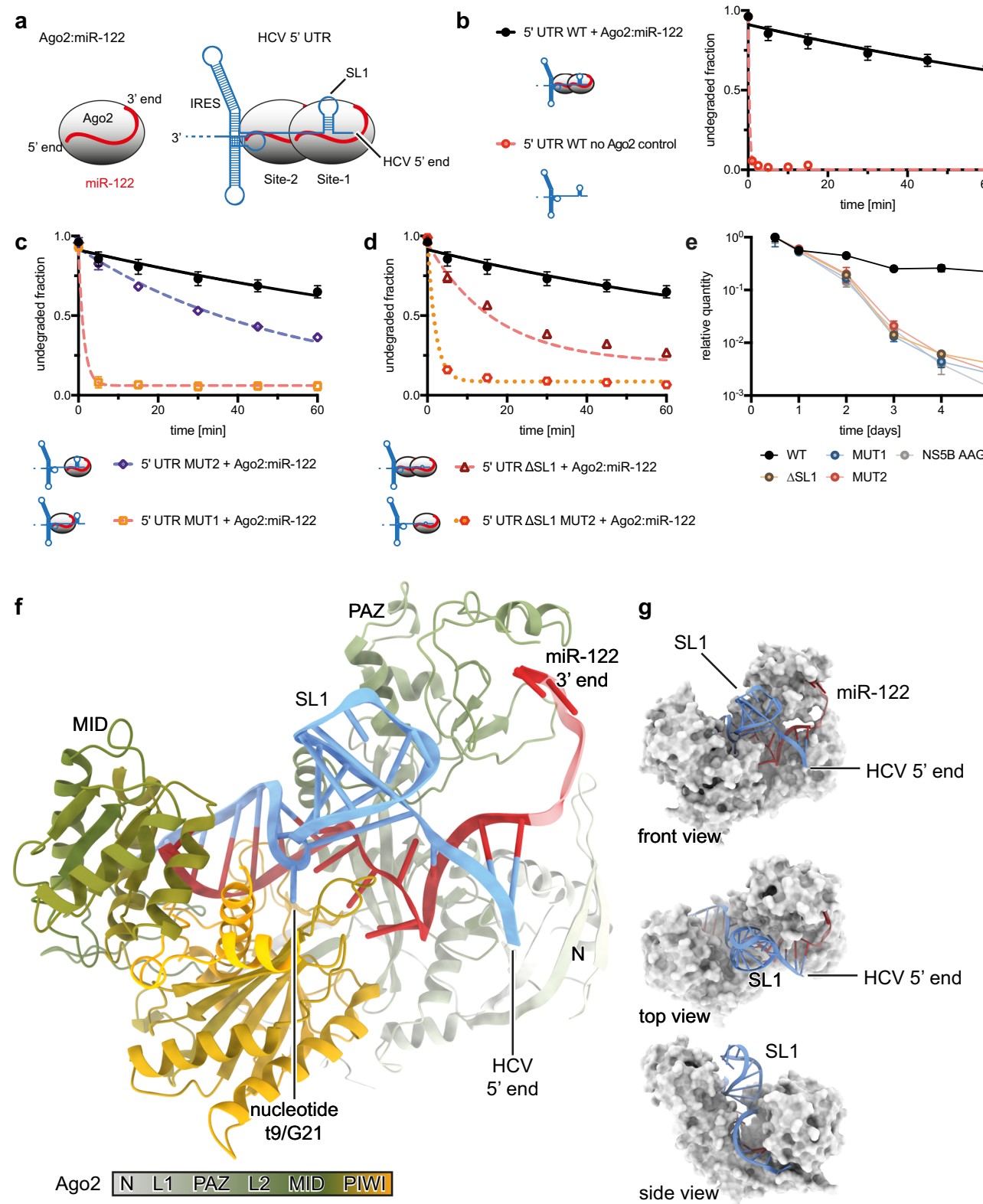

mutants. To distinguish between replication and RNA stability effects, we prepared double mutant genomic RNAs, containing NS5B GDD > AAG in addition to MUT1 or ΔSL1. Viral RNA degradation immediately the following electroporation is driven by multiple mechanisms, including exosome attack of the viral 3' end[20]. Nonetheless, RT-PCR revealed a small but significant effect of the 5' terminal mutations on RNA stability (Supplementary Fig. 1g). The combined results demonstrate the SL motif at Site-1

is essential to HCV fitness in Huh-7.5.1 cells, and that SL1 is necessary for Ago2-mediated protection of the viral 5' UTR from XRN1, especially in the absence of Ago2 occupancy at Site-2.

**Crystal structure of the Ago2:miR-122:HCV Site-1 complex.** As features such as SL1 have not yet been recognized in cellular miRNA targets, we investigated the structure of the Ago2:miR-122:HCV Site-1 ternary complex. We crystallized recombinant

**Fig. 1 Ago2 occupancy at Site-1 protects HCV RNA from degradation and crystal structure of the protective complex. a** Cartoon schematic of Ago2:miR-122 complex (left). Schematic of the HCV 5' UTR with Ago2-miR-122 complexes bound to miR-122 binding Site-1 and Site-2 at the 5' end of the genome, upstream of the IRES (right). Stem-loop 1 (SL1) is embedded within miR-122 Site-1. **b** Degradation of HCV 5' UTR wild type (WT) RNA by XRN1 in the presence or absence of Ago2:miR-122 over time. Data from 3 independent replicates with error bars representing SEM. **c** Degradation of HCV 5' UTR RNAs by XRN1 with mutations that disrupt miR-122 binding at Site-1 (MUT1) or Site-2 (MUT2) compared to WT 5' UTR data from 1B. Data from 3 independent replicates with error bars representing SEM. **d** Degradation of HCV 5' UTR ΔSL1 RNA and HCV 5' UTR ΔSL1 MUT2 RNA by XRN1 compared to WT data from 1B. Data from 3 independent replicates with error bars representing SEM. **e** Relative abundance over 5 days of viral RNA after electroporation into Huh-7 cells. Mutations (WT, MUT1, MUT2, ΔSL1, NS5B GDD > AAG) indicated. Data from 3 independent replicates with error bars representing SEM. **f** Crystal structure of Ago2:miR-122 bound to HCV Site-1 RNA. Ago domains are coloured from grey (N-terminal), through olive (PAZ and MID), to gold (PIWI). miR-122 in red, HCV RNA in blue. Stem-loop 1 (SL1) extends from the center of the complex. **g** Front, top and side views with Ago2 in surface representation (grey).

## Table 1 Crystallographic Data and Refinement Statistics.

| PDB ID | 7KI3 |
|---|---|
| Wavelength (Å) | 1 |
| Resolution range (Å) | 99.23–3.0 (3.107–3.0) |
| Space group | P 21 21 21 |
| Unit cell dimensions (Å) | 87.3133, 112.963, 207.684 |
| Total reflections | 529882 (50608) |
| Unique reflections | 41939 (4134) |
| Multiplicity | 12.6 (12.2) |
| Completeness (%) | 99.96 (99.98) |
| Mean I/sigma(I) | 12.52 (1.58) |
| Wilson B-factor | 56.68 |
| R-merge | 0.4131 (1.147) |
| R-meas | 0.4308 (1.198) |
| R-pim | 0.1208 (0.3412) |
| CC1/2 | 0.885 (0.537) |
| CC* | 0.969 (0.836) |
| Reflections used in refinement | 41923 (4133) |
| Reflections used for R-free | 2058 (203) |
| R-work | 0.2298 (0.2972) |
| R-free | 0.2686 (0.3352) |
| CC(work) | 0.876 (0.635) |
| CC(free) | 0.884 (0.681) |
| Number of non-hydrogen atoms | 14417 |
| macromolecules | 14414 |
| ligands | 3 |
| Protein residues | 1587 |
| RMSD bonds (Å) | 0.005 |
| RMSD angles (°) | 1.05 |
| Ramachandran favored (%) | 94.22 |
| Ramachandran allowed (%) | 5.78 |
| Ramachandran outliers (%) | 0 |
| Rotamer outliers (%) | 2.57 |
| Clashscore | 4.7 |
| Average B-factor | 64.01 |
| macromolecules | 64.01 |
| ligands | 84.47 |

Numbers in parentheses represent statistics in the highest resolution shell. R-merge = $(\Sigma_{hkl}\Sigma_j |I_{hkl,j} - (I_{hkl})|)/(\Sigma_{hkl}\Sigma_j I_{hkl,j})$, where $(I_{hkl})$ is the mean of related observations of a unique reflection. R-meas = $(\Sigma_{hkl}(n/(n-1)^{1/2}\Sigma_{j=1}^n |I_{hkl,j} - (I_{hkl})|)/(\Sigma_{hkl}\Sigma_j I_{hkl,j})$. R-pim = $(\Sigma_{hkl}(1/(n-1)^{1/2}\Sigma_{j=1}^n |I_{hkl,j} - (I_{hkl})|)/(\Sigma_{hkl}\Sigma_j I_{hkl,j})$. R-work and R-free = $\Sigma\Pi F_o| - |F_c\Pi/\Sigma|F_o| \times 100$ for 95% of recorded data (R-work) or 5% of data (R-free). CC, correlation coefficient; RMSD, root-mean-square deviation.

Ago2 loaded with miR-122 in complex with a synthetic oligo-nucleotide corresponding to a variation of nt 1-29 of genotype 1 HCV. Diffraction data were collected to a resolution of 3 Å (PDB ID: 7KI3, Table 1, and Supplementary Fig. 2). Ago2, with its four globular domains (N-terminal, PAZ, MID and PIWI, with N and PAZ connected by linker L1, and PAZ and MID connected by linker L2) is found in a wide-open conformation cradling the viral RNA, with SL1 prominently extending from between the PAZ and PIWI domains. The viral 5'-terminus is positioned on the surface of the complex, lying against the N-terminal domain (Fig. 1f).

As in previous Ago2-miRNA-target structures[31,32], miR-122 threads through the central cleft of Ago2, with the 5' end bound to the MID domain and the 3' end bound to the PAZ domain (Fig. 1f, g and Supplementary Fig. 3a). HCV nucleotides A22-C28 are paired to the miRNA seed region, and HCV nucleotide A29 is anchored to the target nucleotide 1-A-binding pocket in the Ago2 MID domain[31,33]. The central region of the miRNA threads underneath SL1, to rejoin pairing to the viral RNA in the miR-122 supplementary region (Fig. 2a, and Supplementary Fig. 3b, c). The viral RNA and miR-122 thus form a three-way junction (Fig. 2a), with the HCV-paired seed region, SL1, and HCV-paired supplementary region intersecting in the center of Ago2, cradled within a structure termed the central gate[34]. Intriguingly, the viral nucleotide opposite miR-122 nucleotide 9 (target 9, t9; viral nucleotide G21), positioned between the seed region and SL1, is flipped out from the seed-paired duplex and bound in a previously unrecognized nucleotide-binding pocket on the surface of Ago2 (Supplementary Fig. 3d).

We identified extensive interactions between the HCV RNA and Ago2, including the 5' end of Site-1 (paired to miR-122 supplementary region), the region around SL1, the region around t9/G21, and the Site-1 3' end (Fig. 2b). Sequence alignments reveal C2 and C3 in the supplementary-paired region, the seed-paired region, and a purine at position t9 are conserved features of HCV genotype 1 isolates as well as HCV sub-genotypes (Fig. 2c). Sequences corresponding to SL1 appear less conserved at a first glance. The apical loop shows very poor conservation, and, indeed, is exposed to the solvent, with no specific contacts with Ago2 (Fig. 1g). However, all genotypes are predicted to form a hairpin structure similar to SL1 with an overall mean folding energy of −11.3 kcal/mol (Supplementary Fig. 4a). We conclude that the Ago2:miR122:HCV Site-1 structure presented here is likely representative of all known HCV strains.

**Ago2:miR-122 masks the HCV 5' end from exonucleolytic attack.** The HCV RNA 5' end is held to the Ago2:miR-122 complex through base-pairing interactions to the supplementary region of miR-122 (Fig. 2d). The supplementary pairing was previously shown to be important for HCV RNA accumulation in Huh-7 cells[35]. Indeed, a Site-1 mutant RNA, in which supplementary pairing was disrupted, was rapidly released from the Ago2:miR-122 complex (Fig. 2e) and, in the presence of excess Ago2:miR-122, degraded by XRN1 8-times faster than a WT Site-1 RNA (Fig. 2f). XRN1 resistance also decreased with shortened miR-122 forms (i.e. shorter 3' isomiRs) (Fig. 2g), which promote weaker supplementary interactions[36,37]. These data demonstrate supplementary pairing is essential for protecting the HCV RNA from XRN1, and may explain why the accumulation of HCV RNA in cultured cells requires longer miR-122 isoforms[38].

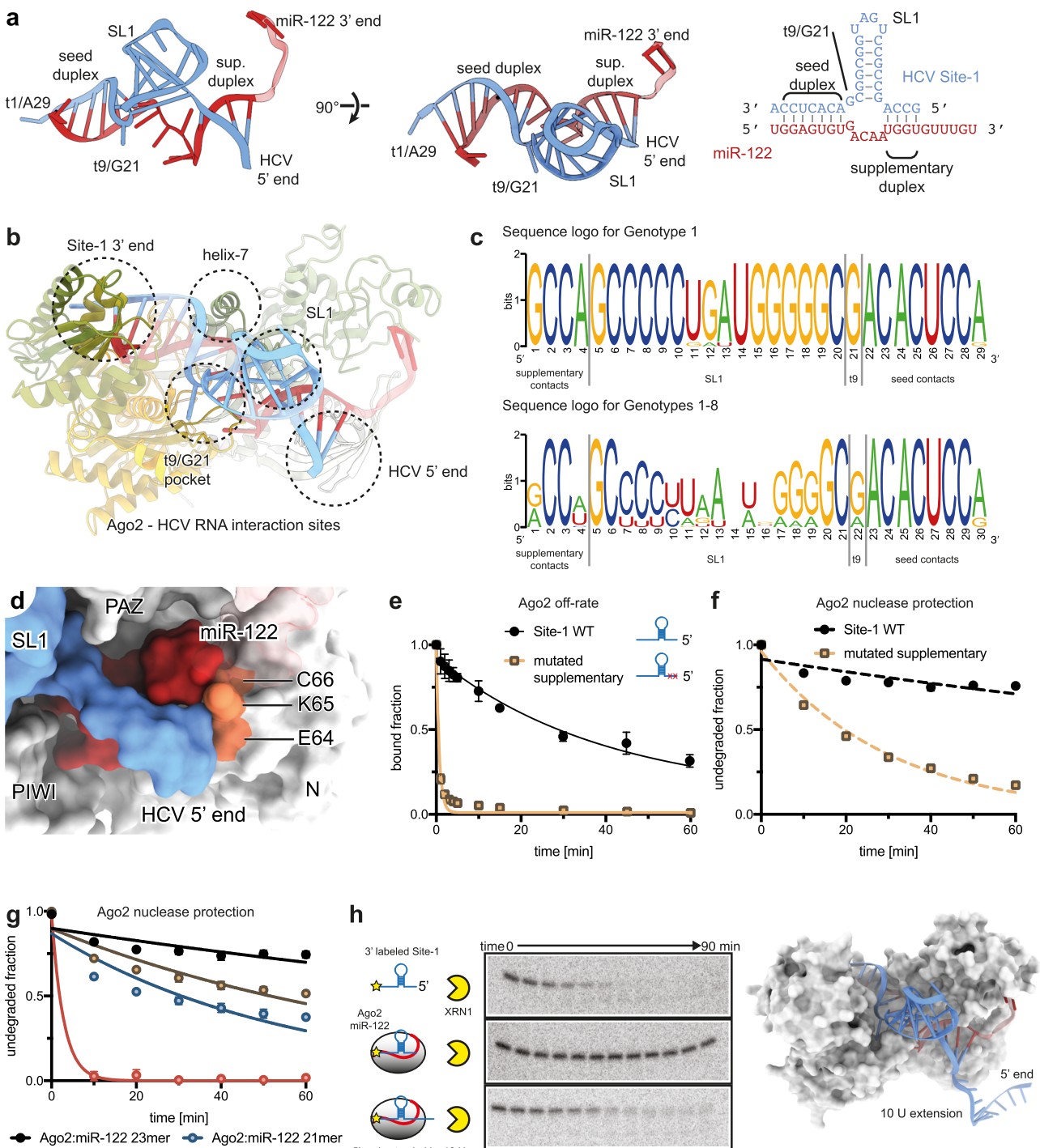

**Fig. 2 Ago2 shields the HCV 5' end from enzymatic attack. a** 3D structures (left) and schematic representation (right) of the miR-122:Site-1 three-way junction, composed of the seed duplex, SL1, and the supplementary duplex. **b** Cartoon representation of Ago2 with major HCV RNA contacts highlighted (determined by PDB PISA[50]). **c** Multiple sequence alignment of HCV genotype 1 subtype sequences (top) and genotype 1-8 subtype sequences (bottom). The total height of the letters at each position corresponds to the conservation at that position, while the height of the letters within the stack corresponds to their relative frequency. Supplementary contacts, SL1, t9, and seed contacts are indicated. **d** Surface representation showing closeup of the HCV RNA (blue) 5' end bound to miR-122 (red) and positioned against the N-terminal domain residues E64, K65, and C66 (orange). **e** Ago2:miR-122 off-rate experiment using Site-1 WT and Site-1 with supplementary-pairing region mutated (GCCA > GAAA). Data shown as means of 3 independent replicates with error bars representing SEM. **f** XRN1 mediated degradation of Site-1 WT and with mutated supplementary-pairing region RNAs in the presence of Ago2:miR-122. Data shown as means of 3 independent replicates with error bars representing SEM. **g** XRN1 mediated degradation of WT Site-1 in presence of Ago2 loaded with indicated miR-122 3' isomiRs, or no Ago2. Data shown as means of 3 independent replicates with error bars representing SEM. **h** Polyacrylamide gels showing XRN1 mediated degradation of Site-1 RNA alone (top), Site-1 with Ago2:miR-122 (middle), and Site-1 containing a 5' $U_{10}$ extension with Ago2:miR-122. Images are representative gels of triplicate experiments. Ago2:miR-122:Site-1 structure with $U_{10}$ extension modelled (right) to illustrate potential 5' end exposure.

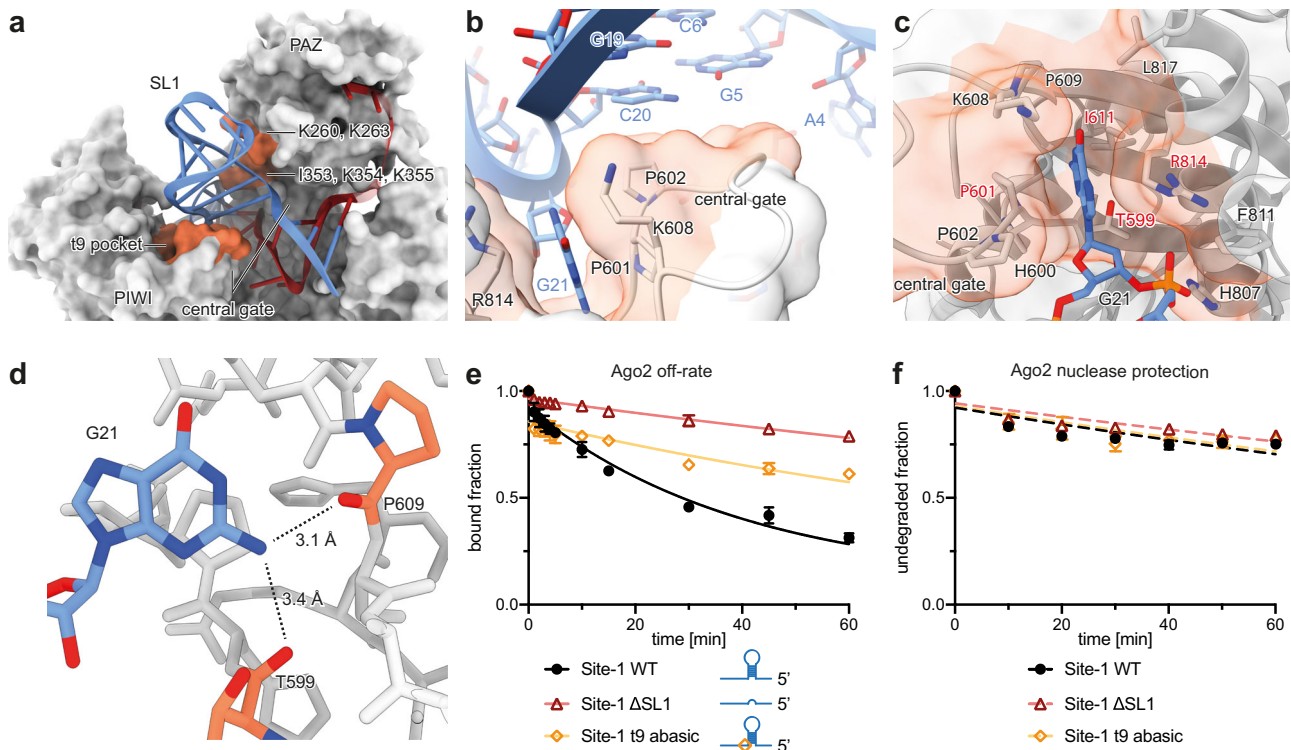

**Fig. 3 SL1 and G21/t9 hold Ago2 in an open conformation. a** Detailed interactions between Ago2 and SL1. SL1 inserts between the PAZ and PIWI domains, on top of the PIWI loop of the central gate, a structure that antagonizes continuous base pairing to the miRNA central region. Ago2 surface contacting the viral RNA is shaded in orange. The interface with the PAZ domain is formed by two rows of residues. **b** Close up view showing the PIWI domain central gate loop inserts under SL1. Note this loop also contributes to the G21/t9 binding pocket. Ago2 surface contacting the viral RNA is shaded in orange. **c** Close up of PIWI domain G21/t9-binding pocket, shown with Ago2 in cartoon representation with transparent surface, and interface residues as sticks. Main contributors to the nucleobase interface labeled in red. Ago2 surface contacting the viral RNA is shaded in orange. **d** Side view of G21 in the hydrophobic pocket. Functional groups within H-bond distance of the 2-amino moiety indicated by dashes. **e** Release of Site-1 WT, Site-1 ΔSL1, and Site-1 t9 abasic from Ago2:miR-122 over time. Data shown as means of 3 independent replicates with error bars representing SEM. **f** Degradation of Ago2:miR-122 bound Site-1 WT, Site-1 ΔSL1, and Site-1 t9 abasic over time in presence of XRN1. Data shown as means of 3 independent replicates with error bars representing SEM.

The HCV 5′ end is pinned against a loop in the Ago2 N-terminal domain (residues E64, K65, and C66) (Fig. 2d). We, therefore, hypothesized supplementary interactions protect the viral RNA by holding the 5′ end in a position inaccessible to enzymatic attack. Alternatively, we imagined the intact ternary complex may create a physical barrier impassible by XRN1. This mechanism would be reminiscent of XRN1-resistant pseudoknot structures in flavivirus RNAs known to stop the processive 5′-3′ exonuclease[39].

To distinguish between these possibilities, we designed a Site-1 RNA variant with an extended 5′ tail of 10 U residues (Fig. 2h). Our rationale was the extended 5′ tail would ensure XRN1 has access to the viral RNA 5′ end when Ago2:miR-122 is bound. We radiolabeled the WT and tailed Site-1 RNAs at the 3′ ends so that degradation intermediates of the 5′-3′ exonuclease could be visualized. Control experiments revealed Ago2:miR-122 strongly protected the WT Site-1 RNA from XRN1. In contrast, Ago2:miR-122 did not protect the 5′-tailed construct, which was rapidly degraded. The disappearance of the tailed RNA was not accompanied by the accumulation of degradation intermediates. Thus, the Site-1 motif cannot stop XRN1 once it engages the viral RNA. We conclude Ago2:miR-122 protects HCV RNA by sterically shielding the 5′ end from enzymatic attack.

**Site-1 elements SL1 and t9 hold Ago2 in an open conformation.** SL1 is wedged between the Ago2 PAZ and PIWI domains,

holding open the central gate, a structure in Ago2 that discourages pairing to nucleotides in the miRNA central region (guide (g) nucleotides g9–g12) (Fig. 3a)[36]. On the PAZ side, the interface is primarily formed by two rows of positively charged residues: K260 and K263 of the PAZ domain, and K354 and K355, which reside on the PAZ side of the central gate along with I353, which stacks below the ribose of HCV RNA C6. On the PIWI side, the PIWI loop inserts into the middle of the HCV-miR-122 three-way junction, making contact with nucleobases at the base of SL1 (Fig. 3b).

Adjacent to SL1, HCV nucleotide t9/G21 binds a pocket formed by the PIWI loop of the central gate. The t9/G21 purine nucleobase inserts between P601 and the aliphatic portion of R814 (Fig. 3c). While all HCV genotypes possess a purine base at this position, the over-representation of a G (Fig. 2d) might be explained by hydrogen bonds between the G21 2-amino moiety and backbone carbonyls of T599 and P609 (Fig. 3c). PIWI domain residues forming the pocket are highly conserved among metazoan miRNA-loading Ago proteins (Supplementary Fig. 5).

To examine the contribution of SL1 and t9 to HCV RNA binding we measured release rates ($k_{off}$) of RNAs corresponding to wild type Site-1 (WT Site-1), Site-1 with SL1 replaced by an $A_4$ bridge (Site-1 ΔSL1), and a Site-1 with an abasic nucleotide at position t9 (Site-1 t9 abasic) (Fig. 3e). Surprisingly, release of the ΔSL1 and t9 abasic RNAs from Ago2:miR-122 was ~6-fold and ~3-fold slower than $k_{off}$ of the WT Site-1 RNA, respectively (Fig. 3e). Thus, despite the numerous contacts described above,

SL1 and t9/G25 are both detrimental to Site-1 binding compared to a Site-1 mutant RNA with an unpaired $A_4$ bridge between seed and supplementary paired regions. We suspect these differences may reflect tension in the Ago2:miR-122 complex, which must adopt an extended conformation to engage SL1 (Fig. 3a).

We also examined Ago2:miR-122-mediated XRN1 protection and found ΔSL1 and t9 abasic Site-1 constructs behaved almost identically to WT Site-1 (Fig. 3f). This observation was puzzling, considering the differences in $k_{off}$ described above (Fig. 3e) and earlier results showing removal of SL1 from the full HCV 5' UTR reduced protection from XRN1 (Fig. 1e).

**Extensions 3' of Site-1 have a deprotective effect that is counteracted by SL1.** Our experiments lead to an apparent contradiction: SL1 removal clearly reduces protection of the HCV 5'UTR from XRN1 by Ago2:miR-122 (Fig. 1d), and yet SL1 removal does not inhibit binding to the 5' UTR (Supplementary Fig. 1f) and, in fact, decreases $k_{off}$ of Ago2:miR-122 dissociating from an RNA corresponding to Site-1 alone (Fig. 3e).

The first clue for resolving this conundrum was the observation that SL1 removal had no effect on protection from XRN1 when using a short RNA construct limited to Site-1 (Fig. 3f), indicating RNA downstream of Site-1 influences our results. To explore this possibility, we made Site-1 RNA constructs with an extended 3' tail (Fig. 4a). We chose a tail sequence corresponding to mutated HCV Site-2 (Supplementary Fig. 1b), which does not stably bind wild type Ago2:miR-122 (Supplementary Fig. 1b, c). In contrast to results with RNAs limited to Site-1 (Fig. 3f), XRN1 degradation of the extended Site-1 ΔSL1 RNA was >16-fold faster than degradation of extended Site-1 WT RNA (Fig. 4b). Similarly, XRN1 degradation of the extended Site-1 t9 abasic was ~2.5 times faster than the extended Site-1 WT RNA (Fig. 4b). Thus, RNA downstream of Site-1 diminishes the ability of Ago2:miR-122 to protect the HCV 5' end from attack by XRN1 in the absence of the SL1 motif.

The Site-1 motif contributes to nuclease protection by reducing Ago2 lateral diffusion

To understand how SL1 contributes to protection from XRN1 we reexamined the structure of the Ago2:miR-122:Site-1 complex. We noticed that by propping open the Ago2 central gate (Fig. 4c), SL1 also props up helix-7, a structural element that enables Ago2 to make and break base-pairing interactions in miRNA seed region, as well as a shuttle between binding sites on a target RNA without physically dissociating from the target RNA molecule, a process termed lateral diffusion[26,27]. Compared with the structure of Ago2 engaged with a target with seed and supplementary pairing, helix-7 in the Site-1 structure makes fewer contacts to the seed-paired RNAs (Fig. 4d). We, therefore, hypothesized SL1 and t9/G21 might enhance the protective effect of Ago2:miR-122 by counteracting lateral diffusion.

To test this model, we first determined how 3' extensions impact interactions with Ago2. In all cases, 3' extended constructs exhibited a ~2-fold reduced $k_{off}$ compared with their unextended counterparts (Fig. 4e). Thus, the 3' extensions moderately stabilize interactions with bound Ago2:miR-122 molecules. Importantly, the rate of dissociation from the 3' extended ΔSL1 RNA was >20-times slower than the rate of degradation by XRN1 (Fig. 4b) under similar conditions, indicating that most degradation occurs while Ago2 is physically associated with the RNA.

To further test the model, we determined whether protection of the extended Site-1 ΔSL1 construct could be restored by occupying the RNA 3' extension with a second Ago2 protein. Our rationale was that binding a second Ago2:miRNA should sterically prevent Ago2:miR-122 molecules at Site-1 from laterally diffusing onto the 3' tail. Indeed, the addition of Ago2 loaded

with a guide complementary to the 3' extension completely restored protection from XRN1, increasing the RNA half-life 25-fold (Fig. 4f). These results are similar to our early observation that removal of SL1 in combination with mutation of Site-2 (ΔSL1,MUT2) abolished protection of the HCV 5' UTR from XRN1 (Fig. 1d).

If we suppose XRN1 acts by attacking viral RNA molecules released from Ago2:miR-122, the rate of RNA degradation by XRN1 should not exceed Ago2 off-rates in our $k_{off}$ measurements. Indeed, this relationship appears true for RNA constructs restricted to the footprint of Site-1 (compare Fig. 2e, f, and Fig. 3e, f). In contrast, the extended Site-1 ΔSL1 and extended Site-1 t9 abasic RNAs are degraded significantly more rapidly than the $k_{off}$ would seemingly allow, indicating the presence of an additional state in which the 5' end is accessible but Ago2:miR-122 is still bound. We suggest this additional state arises from the transient movement of Ago2:miR-122 onto nearby regions of the RNA via lateral diffusion, which can be counteracted by SL1 physically locking Ago2 on the HCV RNA 5' end.

**SL1 directly protects the 5' end from phosphatase.** We also tested the effect of SL1 on degradation by an enzyme other than XRN1, commercial calf intestine alkaline phosphatase (CIP). Dephosphorylation experiments analogous to the XRN1 experiments described above showed the 5' phosphate of extended Site-1 WT RNA was protected from CIP by Ago2:miR-122, with a half-life of >550 min (Supplementary Fig. 6a). For extended Site-1 ΔSL1 the half-life was reduced to 14 min, revealing SL1 is essential for robust protection from CIP. In contrast to results with XRN1, occupying the 3' MUT2 extension with Ago2:miR-122-mut only moderately restored protection to the extended ΔSL1 RNA, increasing the half-life by 40% (to 20 min) (Supplementary Fig. 6a). Similarly, SL1 was necessary for full protection of the minimal Site-1 RNA constructs (lacking 3' extensions) by Ago2:miR-122 (Supplementary Fig. 6b). We conclude that, even in the absence of downstream RNA, SL1 has a direct effect on protection by Ago2:miR-122 against attack by the phosphatase.

**Ago2 and PCBP2 form a mixed complex on the HCV 5' tail.** The genomes of ssRNA viruses act as templates for both protein translation and RNA replication, each necessary to make new virions. As translation proceeds 5' to 3' and the RNA-dependent RNA polymerase replicates 3' to 5', regulation is required[13]. For HCV, one proposed mechanism to alternate translation and replication involves Ago2:miR-122 competing for binding at the viral 5' end with another host protein, PCBP2[15]. We tried to replicate these findings in our reconstituted system, using the 5'-terminal two-site RNA construct from experiments described above. Analysis by EMSA revealed a PCBP2-induced shift (Fig. 5a, lanes 1 and 2), confirming PCBP2 recognizes the HCV 5' tail. To our surprise, including increasing amounts of Ago2:miR-122 with PCBP2 (Fig. 5a, lanes 3 to 5) diminished the shifted band from PCBP2 alone, and created a super-shift, running between the two bands created by Ago2:miR-122 alone (Fig. 5a, lane 6). This result indicates PCBP2 and Ago2:miR-122 can simultaneously occupy the HCV 5' tail, forming a mixed complex.

To determine how each miR-122 binding site contributes to the mixed complex, we repeated the experiment with constructs possessing miR-122 binding mutations at either Site-1 (MUT1) or Site-2 (MUT2). Both mutant RNAs bound PCBP2 comparably to the WT RNA (Fig. 5a, compare lanes 2, 8, and 14). For MUT1, a faint super-shift was visible when adding high amounts of Ago2:miR-122 to PCPB2, but the PCBP2-only shift remained the prominent band (Fig. 5a, lanes 9–11). In contrast, the MUT2 RNA formed a single super-shifted band as observed in WT for

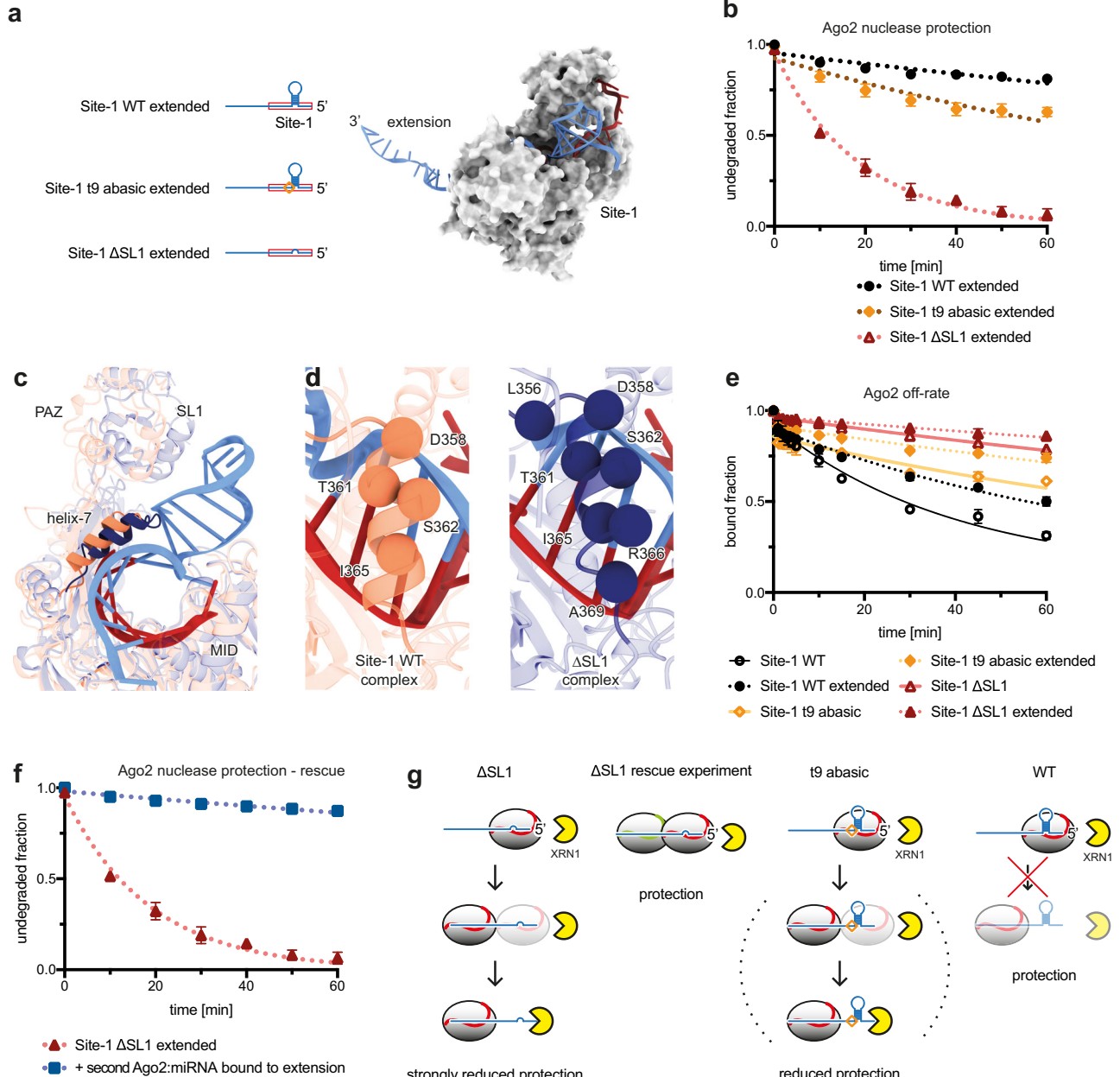

**Fig. 4 SL1 protects from XRN1 by attenuating Ago2 lateral diffusion. a** Cartoons of extended Site-1 WT, ΔSL1 and t9 abasic RNA constructs (blue). Red box indicates Site-1. Model of the extended Site-1 WT (blue ribbon) in complex with Ago2:miR-122, to illustrate the relative size of the 3′ extension. **b** XRN1 mediated degradation of 3′ extended Site-1 WT, t9 abasic, and ΔSL1 in the presence of Ago2:miR-122 over time. Data are shown as means of 3 independent replicates with error bars representing SEM. **c** PAZ domain and helix-7 positions (relative to MID-PIWI lobe) in structures of Ago2:miR-122 bound to HCV Site-1 (coral), or seed plus supplementary paired RNA (dark blue, PDB code 6N4O). **d** Comparison of the Ago2 helix-7-target RNA contacts (spheres centred at Cα atoms in contacting residues) for the Site-1, and seed plus supplementary paired RNA. The miRNA strand is coloured red and the target strand blue (only seed base pairs shown). **e** Release of Site-1 WT, ΔSL1, and t9 abasic Site-1 RNAs, with and without 3′ extensions, from Ago2:miR-122 over time. Data are shown as means of 3 independent replicates with error bars representing SEM. **f** XRN1 mediated degradation of extended Site-1 ΔSL1 RNA in presences of Ago2:miR-122-wt alone or in combination with Ago2:miR-122-mut (targeting the 3′ extension). Data are shown as means of 3 independent replicates with error bars representing SEM. **g** Cartoon representation of the proposed mechanism. Ago2:miR-122 transiently diffuses onto 3′ extensions in the absence of SL1, allowing XRN1 access to the HCV RNA 5′ end (left). The presence of a second Ago2:miRNA can prevent lateral diffusion, re-establishing protection (middle-left). Ago2:miR-122 also laterally diffuses from a Site-1 with a t9 abasic nucleotide, albeit at a reduced rate compared with the ΔSL1 construct, again enabling XRN1(middle-right). The full motif at Site-1 prevents Ago2:miR-122 lateral diffusion, and the target RNA is protected (right).

Ago2 with PCPB2 (Fig. 5a, lanes 15–17). Testing short RNAs corresponding to the individual sites revealed no PCBP2 binding to Site-1, and apparent competition between PCBP2 and Ago2 at Site-2 (Supplementary Fig. 7a, b). These results suggest PCBP2 binding to the 5′ tail is compatible with Ago2 occupying Site-1, and competitive with Ago2 for binding Site-2.

Finally, we tested whether PCBP2 could take the place of Ago2 at Site-2 to prevent 5′ end attack by XRN1 in the absence of SL1.

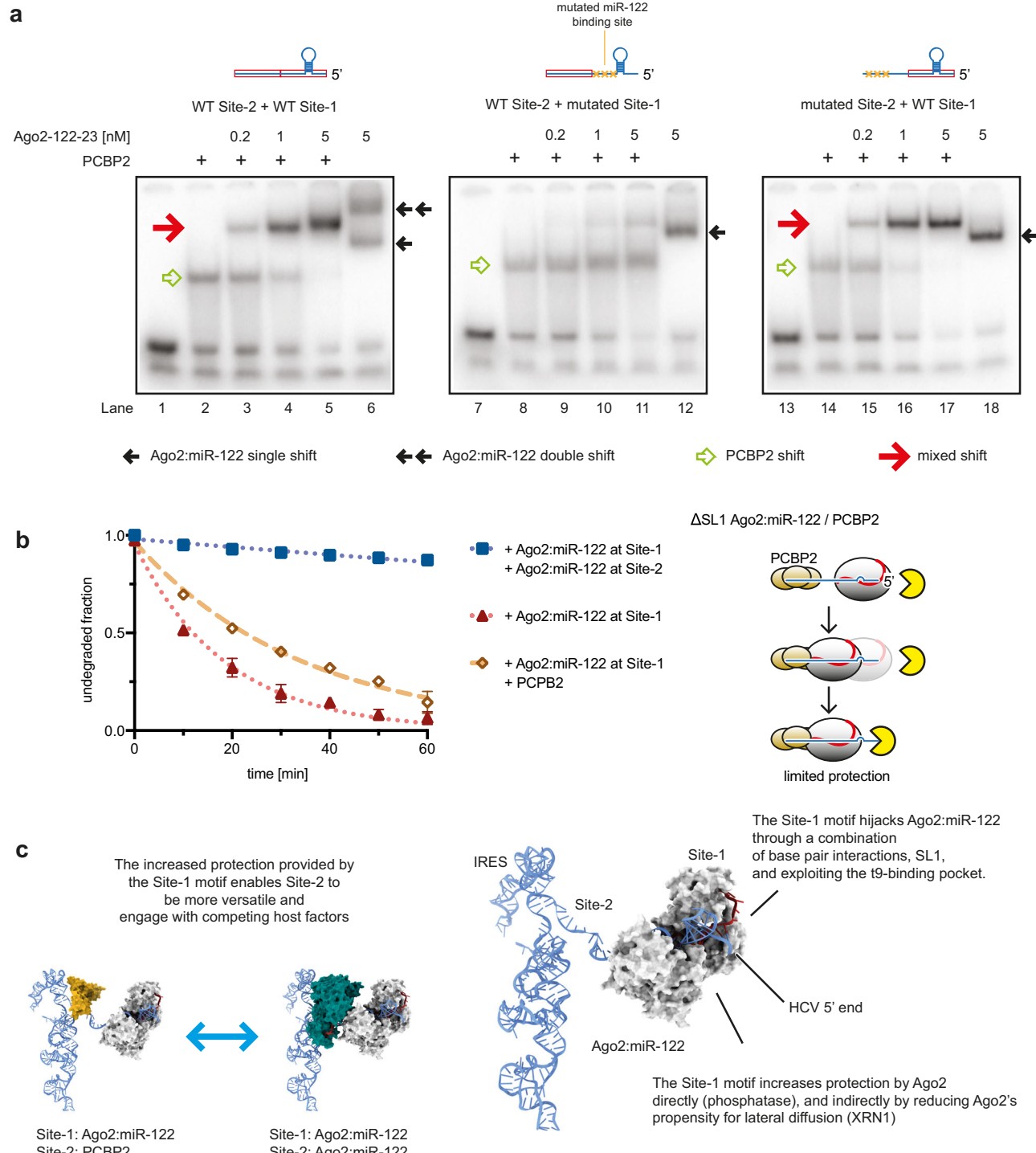

**Fig. 5 A mixed Ago2:miR-122:PCBP2 complex can form on the HCV 5′ tail. a** EMSA of HCV 5′ tail RNAs by PCBP2 and/or Ago2:miR-122. WT, MUT1, and MUT2 RNAs were incubated with PCBP2 (100 nM), and/or Ago2:miR-122 (concentrations indicated), or both prior to running the gel. Lanes 1, 7, and 13 show free RNA for WT, MUT1, or MUT2. Schematic of HCV 5′ tail RNAs (blue) with miR-122 binding sites as red boxes and mutated sites indicated by orange X's, show at the top. The experiment was repeated twice and a representative gel is shown. **b** XRN1 mediated degradation of Site-1 ΔSL1 MUT2 RNA in presence of Ago2:miR-122-wt, Ago2:miR-122-wt and PCBP2, or Ago2:miR-122-wt and -mut. Data shown as means of 3 independent replicates with error bars representing SEM. The cartoon (right) describes the proposed mechanism, with PCBP2 occupancy at Site-2 only partially reducing Ago2:miR-122 lateral diffusion from Site-1. **c** Model of Ago2:miR-122 bound to Site-1 with HCV IRES shown for scale (PDBID: 5a2q). Ago2 occupancy at Site-1 may enable a molecular switch in which Site-2 occupancy alternates between Ago2:miR-122 and PCBP2 (represented as 3 KH domains, PDBID: 2p2r).

Compared to Ago2:miR-122 at Site-1 ΔSL1 alone, including PCBP2 increased the half-life of the RNA significantly (from 13 min to 24 min) (Fig. 5b), but far less than the 326 min observed for two Ago2 molecules binding in tandem (Fig. 4f).

These data provide additional evidence for the presence of a molecular switch between Ago2 and PCBP2, but compared to the previous model, we find that this switch occurs specifically at Site-2. Furthermore, Site-1 has the ability to remain occupied by Ago2

independently of the state of Site-2 (Fig. 5a, c). Occupancy of Ago2:miR-122 at Site-1 may be beneficial for masking the viral 5' end and maintaining the integrity of the genomic RNA. Notably, the mixed complex containing Ago2:miR-122 and PCBP2 cannot provide full nuclease protection in absence of SL1 (Fig. 5b), that may further explain the importance of the SL1 motif during the viral life cycle.

## Discussion

Here we dissected the two HCV miR-122 sites with respect to protection from XRN1, finding that Site-1 is essential, and Site-2 has a synergistic effect. Importantly, the conserved SL1 within Site-1 is necessary for Ago2:miR-122-mediated protection of HCV RNA from XRN1 and CIP. The crystal structure of Site-1 in complex with Ago2:miR-122 revealed a particularly open conformation of Ago2, through the use of multiple RNA-protein interaction sites, including a previously unknown t9 purine-binding site in the Ago2 PIWI domain. We propose Site-1 represents a previously unrecognized RNA motif that combines sequence-specific pairing to the miRNA with the secondary structure of SL1 to lock Ago2 in an open conformation on the HCV RNA 5' end.

Our findings highlight the modular architecture of the HCV 5' UTR. We suggest Site-1 has a primary protective function, which in turn enables versatile roles for Site-2. While Site-1 remains Ago2-bound, Site-2 can bind PCBP2 instead of Ago2, forming a mixed complex. PCBP2 has been proposed to contribute to the regulation of replication versus translation in the viral life cycle[15]. Our data raise the possibility that the switch between replication and translation involves occupancy of either Ago2 or PCBP2, or other RBPs, at Site-2. We propose the existence of a mixed Ago2:PCBP2 complex, which along with the multiple roles for Ago2, might explain seemingly contradictory results indicating miR-122 can both stimulate[16] and inhibit[15] viral protein translation (Fig. 5c). Further studies will be required to determine how Ago2:miR-122, PCBP2, and other host factors interact with the HCV IRES and host translational machinery.

A structure similar to SL1, but lacking supplementary pairing, was also identified in a miR-21 binding site near the polyprotein start codon of the Zika virus genome[11]. Another potentially similar interaction occurs at the 3' end of the BVDV genomic RNA, which has been reported to bind the host let-7 and miR-17[10]. In both cases, the binding of the miRNAs to the viral genomic RNA was beneficial to the virus. We hypothesize that the molecular mechanisms found in HCV, namely using Ago:miRNPs to recognize and stabilize specific RNA folds and compete with other RBPs, may be utilized by these related RNA viruses. Further studies will be required to understand how viral RNA genomes interact with components of the miRNA pathway downstream of Ago2.

Of further interest for miRNA biology is the observation that the SL1 structure does not reduce miRNA-mediated silencing in a luciferase reporter (Supplementary Fig. 8), suggesting HCV Site-1 may represent a previously unrecognized class of structured miRNA-recognition site. Notably, SL1 protrudes beyond the Ago2:miR-122 complex such that it could be extended or elaborated with additional RNA structures without perturbing interactions with Ago2 (Fig. 1g). Thus, SL1-like structures in cellular RNAs could potentially bridge pairing to seed and supplementary regions in sequences far apart in a target RNA's primary sequence, or, taken to the extreme, exist on two individual RNA molecules linked through base pairing. Finally, we suggest Ago dynamics on mRNA targets might contribute to the outcome of target recognition, as our findings show RNA motifs, inside and beyond a target site, and interactions with neighbouring RBPs, modulate Ago2 behaviour.

## Methods

**Oligonucleotides**. Short oligonucleotides and primers were ordered from IDT (Coralville, IA) either desalted (for biochemistry) or HPLC-purified (for crystallography) (Supplementary Table S1).

**Preparation of human Ago2 loaded with specific miRNA guides**. Human Ago2 was expressed in Sf9 cells using the baculovirus system according to a previously published protocol[32]. Pelleted cells were resuspended in lysis buffer (50 mM $NaH_2PO_4$ pH 8, 100 mM NaCl, and 0.5 mM TCEP) and lysed by three passages through an M-110P microfluidizer (Microfluidics). The lysate was clarified by centrifugation at 29000 g for 15 min and initial purification by Ni affinity (Ni NTA agarose, Qiagen) was combined with Micrococcal nuclease (Takara) treatment to remove bound RNA (buffer 50 mM Tris-HCl pH 8, 300 mM NaCl, 20 mM imidazole, 5 mM $CaCl_2$, and 0.5 mM TCEP). Ago2 was loaded with a synthetic miRNA bearing a 5'-terminal phosphate (IDT) and the His tag was removed using Tobacco Etch Virus protease. After removal of the His tag using TEV protease followed by reversed Ni affinity purification, Ago2 loaded with a specific micro-RNA was captured using an antisense oligonucleotide (IDT) and eluted with a DNA competitor (IDT). The protein was passed through Q Sepharose Fast Flow resin (Cytiva) to further remove nucleic acids, and purified using size exclusion on a Superdex 75 16 600 column (Cytiva) on an Äkta Pure FPLC (Cytiva) using 50 mM Tris-HCl pH 8, 1 M NaCl, 0.5 mM TCEP. After final dialysis into 30 mM Tris-HCl pH 8, 100 mM NaCl, 0.5 mM TCEP, protein concentration was determined on Nanodrop OneC (Thermo Fisher Scientific) using a calculated extinction coefficient at 280 nm of 198370 $M^{-1}cm^{-1}$. Aliquots were flash frozen in liquid nitrogen and stored at −80 °C until use.

**Crystallization**. Ago2:miR-122 was diluted to 2 g/l (~20 μM) and mixed with synthetic HCV site 1 RNA (IDT) at 1.2x molar equivalents. Initial crystallization experiments were set up manually using 24-well plates (Hampton) using the hanging drop vapour diffusion method and JB Classic screen (Jena Bioscience). Crystal conditions were optimized iteratively using a combination of manual screening and liquid handler (Mosquito, TTP Labtech). Seeds were prepared using the Seed bead kit (Hampton) and used in the iterative optimization. Final crystallization conditions were 6% w/v PEG8000, 50 mM $BaCl_2$, 100 mM Tris-HCl pH 8, yielding elongated hexagons. The crystals were prepared for freezing by transferring to a solution of 7% w/v PEG8000, 50 mM $BaCl_2$, 100 mM Tris-HCl pH8, and 50% w/v sucrose as cryoprotectant.

**X-ray diffraction data collection and processing**. Diffraction experiments were performed at SSRL 12-2, APS GM/CA 23ID-B, and ALS 5.0.2, where the final data set was collected. Diffraction data were indexed using DIALS[40], then scaled and merged with Aimless[41]. Initial phases were generated by molecular replacement using Phaser[42] with the search model comprising the MID and PIWI domains of Ago2 loaded with an unrelated guide RNA (PDBID: 4w5n). Molecular replacement revealed two proteins in the crystallographic asymmetric unit. Strong electron density was observed for a target RNA paired to the miR-122 seed region, the primary interaction site of Ago2:miRNA RNPs with their targets, comprising nucleotides 2-8 of the miRNA guide. The remaining protein (N, L1, PAZ, and L2 domains) and viral stem-loop1 were built by iterative cycles of refinement and modeling, taking advantage of the non-crystallographic symmetry, using Phenix.refine[43], manual adjustments to the model in Coot[44], and real-space refinement of the RNA using ERRASER[45].

**Target RNA labeling**. Synthetic RNA oligonucleotides (IDT) were radiolabeled at the 5'-end using γ-phosphate $^{32}P$-labeled ATP (Perkin Elmer) and T4 poly-nucleotide kinase (NEB), or at the 3'-end using α-phosphate $^{32}P$ pCp (Perkin Elmer) and T4 RNA ligase (Thermo Fisher). Labelled RNAs were purified by denaturing polyacrylamide gel followed by gel extraction and ethanol precipitation. RNA concentration was determined by absorption at 260 nM on a Nanodrop OneC (Thermo Fisher Scientific) using extinction coefficients calculated by the oligo tool from the IDT website (www.idtdna.com).

$k_{off}$ **measurements**. Experiments to measure release of target RNAs from the Ago2:miR-122 complex were performed according a previously published protocol[27].

Ago2:miR-122 (2 nM) was incubated with radiolabeled target (0.1 nM) in reaction buffer (30 mM Tris-HCl pH 8.0, 0.1 M KOAc, 2 mM Mg(OAc)2, 0.5 mM TCEP, 0.005% NP-40) with a total volume of 750 μl and incubated at room temperature for 45 min to reach equilibrium. Protein-bound and free RNAs were separated by passing samples through a Protran nitrocellulose membrane (Amersham, GE Healthcare Life Sciences) and a Hybond N + nylon membrane (Amersham, GE Healthcare Life Sciences) using a dotblot apparatus. Initial (zero) time point was taken by applying 50 μl to the apparatus (with vacuum) and washing with 50 μl wash buffer (30 mM Tris-HCl pH 8.0, 100 mM KOAc, 2 mM Mg(OAc)$_2$, 0.5 mM TCEP). The reaction time course was then initiated by the addition of 7 μl of unlabeled target RNA (100 μM final concentration) to the sample. 50 μl aliquots were passed through the dotblot apparatus at various times, with each time point followed by 50 μl wash buffer. After air drying, the membrane strips were exposed to phosphor screens (GE

Healthcare Life Sciences) for visualization. Screens were imaged on a Typhoon phosphorimager (GE Healthcare Life Science) and the signal was quantified using ImageQuant (GE Healthcare Life Sciences). Dissociation rates were calculated in Prism (GraphPad Software), using a one-phase decay fit, with Y0 set to 1 and the plateau set to 0.001.

**XRN-1 assays**. Recombinant *Kluyveromyces lactis* XRN-1 was expressed in BL21 *Escherichia coli* with a plasmid gifted from the Tong lab[46], and purified by Ni-NTA affinity chromatography followed by size exclusion chromatography on an Äkta Pure FPLC using a Superde x 75 column (Cytiva) as before.

Ago2:miR-122 (100 nM) was incubated with radiolabeled target RNA (1 nM) in reaction buffer (30 mM Tris-HCl pH 8.0, 0.1 M KOAc, 2 mM $Mg(OAc)_2$, 0.5 mM TCEP, 0.005% NP-40, 0.05 g/l BSA, 0.02 U/μl RNase inhibitor (NEB)) in a total volume of 126 μl at room temperature for 45 min prior to starting the XRN-1 reaction.

A timepoint 0 was taken by removing (9 μl) to a fresh tube containing 20 μl of formamide loading buffer. The reaction was started by the addition of 13 μl XRN-1 at 30 nM (final concentration of 3 nM), and 10 μl aliquots were removed and pipetted into formamide buffer to stop the reaction at indicated timepoints.

The aliquots were analyzed on a denaturing polyacrylamide gel (15–20% acrylamide, 8 M urea) run in 0.5x TBE. The gels were used to expose phosphor screens (GE Healthcare Life Sciences) for visualization. Screens were imaged on a Typhoon phosphorimager (GE Healthcare Life Science) and signal was quantified using ImageQuant (GE Healthcare Life Sciences). Dissociation rates were calculated in Prism (GraphPad Software), using a one-phase decay fit, with a Y0 set to 1 and the plateau set to 0.

**PCBP2 expression and purification**. The coding sequence for PCBP2 was amplified from HEK293T cDNA and cloned into a pET23a vector (Novagen) encoding an N-terminal His tag followed by a recognition site for Tobacco Etch Virus (TEV) protease. The recombinant protein was expressed in BL21 *Escherichia coli*, by inducing with IPTG (1 mM) overnight at 16 °C after cultures had reached an $OD_{600}$ of 0.9. Pelleted cells were resuspended in lysis buffer (50 mM $NaH_2PO_4$ pH 8, 100 mM NaCl, and 0.5 mM TCEP) and lysed by three passages through an M-110P microfluidizer (Microfluidics). The lysate was clarified by centrifugation at 29000 g for 15 min and initial purification by Ni affinity (Ni NTA agarose, Qiagen) was combined with Micrococcal nuclease (Takara) treatment to remove bound RNA (buffer 50 mM Tris-HCl pH 8, 300 mM NaCl, 20 mM imidazole, 5 mM $CaCl_2$, and 0.5 mM TCEP). After removal of the His tag using TEV protease followed by reversed Ni affinity purification, the protein was passed through Q Sepharose Fast Flow resin (Cytiva) to further remove nucleic acids, followed by a final purification step using size exclusion on a Superdex 75 10 300 column (Cytiva) on an Äkta Pure FPLC (Cytiva) using 50 mM Tris-HCl pH 8, 1 M NaCl, 0.5 mM TCEP. After final dialysis into 30 mM Tris-HCl pH 8, 100 mM NaCl, 0.5 mM TCEP, the protein was concentrated, supplemented with 20% glycerol, flash frozen in liquid nitrogen, and stored at −80 °C until use.

**PCBP2 competition assays**. 5'-radiolabeled RNAs (IDT) comprising HCV miR-122 binding Site-1 and Site-2 (WT, site 1 mutated, site 2 mutated) were incubated 30 min at room temperature with the indicated concentrations of Ago2 loaded with miR-122 and PCBP2. The reactions were analyzed by native polyacrylamide gel electrophoresis at 4 °C. Gels were dried ~0.5 h on a gel drier at 80 °C to prevent cracking and then used to expose phosphorscreens overnight. The screens were imaged on a Typhoon phosphorimager (GE Healthcare Life Science) and the signal was quantified using ImageQuant (GE Healthcare Life Sciences).

**T7 in vitro transcription for radiolabeling experiments**. DNA templates were prepared by PCR amplification from pUC19 plasmids containing the T7 promoter followed by nt 1-357 of the HCV genome (from H77 deltaE1p7) and a cis-acting HDV ribozyme (to produce homogeneous ends). The wt construct was ordered as a gblock (IDT) and cloned into pUC19, mutants were made using the indicated primers. Transcription reactions were performed overnight at room temperature[47] with an excess of GMP to facilitate 5'-radiolabeling by generating a 5'-monophosphate construct. The template was then digested using RNase-free DNase I, and the RNA was purified using denaturing polyacrylamide gel electrophoresis. Yields were determined on a nanodrop one (Thermo Fisher Scientific), using $A_{260}$ and extinction coefficients obtained from the IDT oligo analyzer tool (www.idtdna.com). The extinction coefficients were calculated assuming single-strandedness as an approximation.

**T7 in vitro transcription for cellular viral RNA stability assays**. DNA templates were prepared by PCR amplification from the plasmid H77 deltaE1p7[29] (wt or mutants prepared with the indicated primers), and the respective mutants and tested by Sanger sequencing. The RNAs were prepared as described above and purified using the RNeasy kit (Qiagen). Yields in ng/μl were determined on a Nanodrop OneC (Thermo Fisher Scientific) using the formula of 40 ng/μl per unit of absorbance at 260 nm.

**Electroporation of Huh-7.5.1 cells**. Electroporation was performed according to a published protocol[48]. Huh-7.5.1 cells were washed twice with DPBS (Thermo Fisher Scientific) and trypsinized. The cells were resuspended in culture medium (DMEM with 0.1 mM nonessential amino acids and 10% fetal bovine serum, Thermo Fisher Scientific) and washed twice with ice-cold DPBS. The second time the resuspended cells were counted, and resuspended to $1.5 \times 10^7$ cells/ml. 5 μg RNA was used for each electroporation. 800 μl cell suspension was added to the prepared RNA. The resulting solution was mixed by pipetting twice and then transferred to an electroporation cuvette. Electroporation was performed on a Gene Pulser (BioRad) according to the published conditions (270 V, 950 μF, 100 Ohm, 4 mm, exponential pulse). Electroporated cells were supplemented with recovery medium with 20% FBS to 10 ml and plated in 6-well plates at 0.3 ml/time point.

**RNA isolation and RT-PCR**. Growth medium was removed and cells were washed once with DPBS. Cells were lysed by the addition of Trizol Reagent (Invitrogen) (0.5 ml per well of a 6-well plate) and total RNA isolated according to the manufacturer's protocol, using glycogen to facilitate precipitation. The RNA was finally resuspended in 10–15 μl RNase-free $H_2O$ and quantified by UV absorbance. Total RNA was reverse transcribed with Superscript III reverse transcriptase (Invitrogen). PCR was performed with Power SYBR Green Master Mix (Applied Biosystems), with primers targeting the IRES[22]. Rlpl2 levels were used as internal standards with the delta Ct method. For comparison, data were fit to one-phase-decay model and decay constants were compared using the extra sum-of-squares F test[22]. PCR primers are listed in Supplementary Table S1.

**Luciferase assays**. Reporter plasmids were prepared by cloning the HCV Site-1 sequence and its mutants (ΔSL1, delta supplementary, ΔSL1 and delta supplementary) into the psiCHECK-2 vector (Promega). HEK293T cells were seeded into 96-well plates. 12 h after seeding, the cells were transfected with 40 ng/well of the indicated plasmid using Lipofectamine 2000 (Thermo Fisher Scientific) according to the manufacturer's protocol. miR-122 was annealed with its passenger strand and transfected 5 h later at the indicated concentrations with Lipofectamine 2000. After 48 h, the medium was removed from the cells, and luciferase activity was assayed using the Dual-Glo Luciferase Assay System (Promega) on a Synergy H1 Hybrid Reader (BioTek).

**Clustal Omega multiple sequence alignment**. Indicated sequences were aligned using the Clustal Omega tool from the website of the EMBL-EBI in Hixton, UK (www.ebi.ac.uk/Tools/msa/clustalo/).

Genbank accession numbers for HCV genotypes were obtained from the website of the International Committee for the Taxonomy of Viruses (https://talk.ictvonline.org). Sequences corresponding to miR-122 Site-1 were obtained from NCBI. The list was manually curated to include one sequence per genotype (using the more recent one in cases where two were published), and to remove any incomplete sequence.

The sequences were aligned using the Clustal Omega tool from the website of the EMBL-EBI in Hixton, UK (www.ebi.ac.uk/Tools/msa/clustalo/). Alignments were performed for all remaining sequences, as well as for the sequences within each genotype. Minimum folding energy for each sequence was calculated using the Vienna RNA package[49].

**Sequence logo generation**. Sequence logos were generated using the online tool from UC Berkeley (weblogo.berkeley.edu/) with the Clustal Omega multiple sequence alignments as input.

**Reporting Summary**. Further information on research design is available in the Nature Research Reporting Summary linked to this article.

## Data availability
Atomic coordinates and diffraction data have been deposited in the Protein Data Bank under accession code 7KI3. The biochemical and cell data generated in this study are provided in the Source Data file. Source data are provided with this paper.

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

## Acknowledgements

Research was funded by fellowships P2EZP3_155572 and P300PA_177860 of the Swiss National Science Foundation and fellowship 15B085 of the Novartis Foundation for Medical-Biological Research to LFRG, NIH grant AI123365 to ML, R35GM127090 to IJM. Beamline 5.0.2 of the Advanced Light Source, a U.S. DOE Office of Science User Facility under Contract No. DE-AC02-05CH11231, is supported in part by the ALS-ENABLE program funded by the National Institutes of Health, National Institute of General Medical Sciences, grant P30 GM124169-01. GM/CA@APS has been funded in whole or in part with Federal funds from the National Cancer Institute (ACB-12002) and the National Institute of General Medical Sciences (AGM-12006). This research used resources of the Advanced Photon Source, a U.S. Department of Energy (DOE) Office of Science User Facility operated for the DOE Office of Science by Argonne National Laboratory under Contract No. DE-AC02-06CH11357. Use of the SSRL, SLAC National Accelerator Laboratory, is supported by the U.S. Department of Energy, Office of Science, Office of Basic Energy Sciences, under contract no. DE-AC02–76SF00515. The SSRL Structural Molecular Biology Program is supported by the DOE Office of Biological and Environmental Research and by the National Institutes of Health, National Institute of General Medical Sciences (including P41GM103393). We thank Erick Giang for technical support.

## Author contributions

L.F.R.G. and I.J.M. designed the study and experiments. M.L. provided expertise, resources, and guidance for HCV cell culture experiments. L.F.R.G. performed the experiments. L.F.R.G. and I.J.M. drafted the paper. All authors edited the paper.

## Competing interests

The authors declare no competing interests.
