## [Peer Review File · Nature Communications]

REVIEWER COMMENTS

Reviewer #1 (Remarks to the Author):

This manuscript describes a crystal structure of the tandem miR-122/Ago2 complex that forms at the 5' end of hepatitis C virus RNA. Elegant structural and biochemical approaches show that the complex protects the viral 5' end from degradation by exonuclease XRN1. Importantly, a penultimate stem loop structure, SL1, in the viral genome was found to dock the RNA/miR-122 complex between the PAZ and PIWI domains of Ago2. This interaction was important to stabilize the viral genome from degradation. This finding sets a paradigm that Ago2/microRNA complexes can be further stabilized by additional stem loop structures. Finally, it was shown that cellular protein PCBP2 can replace the Ago2/miR-122 complex at the distant site 2, suggesting that this interaction may cause the switch from viral translation to viral mRNA translation. This is an outstanding study that reveals novel insights into microRNA/mRNA complexes. Very surprisingly, I have very few comments that could improve the manuscript.

Comments:

1. The loop sequences in SL1 have little effect on viral growth. Can the authors comment in more detail how these sequences interact with the Ago2/miR-122 complex?
2. While PCBP2 has been reported to affect a switch from translation to replication, the effects were modest in full-length viral RNAs (ref. 15). Other RNA binding proteins have been implicated in the switch from translation to replication. In the absence of showing direct effects of PCBP2 on the switch, the authors may tune down or expand their discussion on PCBP2 being important for the switch from translation to replication.

Reviewer #2 (Remarks to the Author):

The Ian MacRae group report on the structure of the 5'-end of HCV RNA bound to the Ago2:miR-122 complex and undertake structure-guided functional studies to deduce mechanistic insights into viral-host interactions that contribute to viral fitness. The structure outlines the binding of the HCV 5'-end containing stem-loop 1 to Ago2:miR-122 at site-1, thereby masking the 5'-end of viral RNA from enzymatic attack by nucleases and phosphatases. The authors identify the pairing between miRNA and HCV RNA, as well as viral RNA-Ago protein contacts, establishing several novel intermolecular interactions. The authors also identify a previously unidentified nucleotide binding pocket that locks Ago2 in a widened conformation and curbs lateral diffusion of Ago2 on target RNAs. They further show that a second Ago2:miR-122 can bind to adjacent site-2 on HCV RNA and protein PCBP2 competes for this site, suggestive of a molecular switch governing partitioning between regulation and translation.

This is a very important paper on how a viral RNA hijacks miRNA targets, with broad implications for the field. Both the structure and follow up functional experiments together with the writeup meet the highest standards. I strongly recommend publication of the paper in its present form after the authors address two minor comments listed below.

Minor comments:

Did the authors attempt to crystallize Ago2:miR122 bound simultaneously to sites-1 and-2 on the 5'-end of HCV RNA, or alternately Ago2:miR122 bound to site-1 and PCBP2 bound to site-2 on the 5'-end of HCV RNA. It may be even possible to include the IRES RNA in the HCV RNA construct and solve the structures by cryo-EM. A comment regarding the feasibility of such an effort should be included in the Discussion.

The later panels appear to be mislabeled in Figure 4 and should be corrected.

Reviewer #3 (Remarks to the Author):

Gebert et al.

A viral RNA structure that hijacks microRNAs

In this manuscript, the authors have further characterized the molecular details of the miR-122 interaction with the 5' end of the HCV genomic RNA. The 5' end of the HCV RNA contains two miR-122 binding sites upstream of an extended secondary structure serving as IRES. Interestingly, site 1 contains a short stem-loop that is not complementary to miR-122 and loops out. The interaction is stabilized by complementarity 3' of the short stem-loop. Gebert et al. have used biochemical approaches and report that deletion of the short stem-loop, as well as the individual miR-122 target site destabilize the HCV RNA. The miR-122-Ago complex at the very 5' end of the HCV RNA protects it from XRN1 exonucleolytic degradation. To unravel the underlying mechanisms, the authors solved the crystal structure of a ternary Ago2-miR-122-HCV (1-29) structure. From the structural data, they conclude that Ago2-miR-122 masks the 5' end and protects it from XRN1-mediated degradation. Furthermore, the short stem-loop holds Ago2 in an open conformation, which is supported by a specific nucleotide on the HCV target site (t9). Interestingly, these motifs prevent Ago2 from lateral diffusion, which is known from 'regular' miRNA target site recognition on host mRNAs. The second miR-122 binding site immediately downstream of the first target site further restricts lateral diffusion and the 5' end is efficiently protected. Finally, the authors report that the translational factor PCBP2 and the Ago2-miR-122 complex can bind simultaneously to the viral RNA allowing for target RNA protection and translation at the same time.

This is a well-written and very clear manuscript. It reports novel biochemical findings in an area that has been studied extensively during the past years. The reported crystal structure provides a clear and convincing model how and why the Ago2-miR-122 complex associates with the 5' end of the HCV RNA. Particularly, the model of restricting lateral diffusion appears appealing. I have only a few minor points that should be considered.

1. The authors propose a novel pocket in Ago2 that accommodates the unpaired target nucleotide t9/G21. Is this really a specific pocket or simply space where the flipped-out nucleotide is positioned? This structure seems to be unique to the HCV target RNA and thus might not be relevant for all other host miR-target interactions. Thus, a positive selection for such a pocket is unlikely. Are there other such target sites known on the human transcriptome?

2. The title is too broad. It is known that the HCV 5' end interacts and hijacks miR-122. This is not new. Furthermore, it only hijacks miR-122 in liver cells. The title should therefore be more specific.

Point-by-point response to the reviewers' comments

REVIEWER COMMENTS

Reviewer #1 (Remarks to the Author):

This manuscript describes a crystal structure of the tandem miR-122/Ago2 complex that forms at the 5' end of hepatitis C virus RNA. Elegant structural and biochemical approaches show that the complex protects the viral 5' end from degradation by exonuclease XRN1. Importantly, a penultimate stem loop structure, SL1, in the viral genome was found to dock the RNA/miR-122 complex between the PAZ and PIWI domains of Ago2. This interaction was important to stabilize the viral genome from degradation. This finding sets a paradigm that Ago2/microRNA complexes can be further stabilized by additional stem loop structures. Finally, it was shown that cellular protein PCBP2 can replace the Ago2/miR-122 complex at the distant site 2, suggesting that this interaction may cause the switch from viral translation to viral mRNA translation. This is an outstanding study that reveals novel insights into microRNA/mRNA complexes. Very surprisingly, I have very few comments that could improve the manuscript.

Thank you for the positive and encouraging feedback.

Comments:

1. The loop sequences in SL1 have little effect on viral growth. Can the authors comment in more detail how these sequences interact with the Ago2/miR-122 complex?

Indeed, the sequence of the loop in SL1 appears to be the least conserved feature in the entire 5' UTR. Regarding the Ago2:miR-122 complex, the loop is fully solvent exposed, with no contact to Ago2 or miR-122. This may explain why loop sequences in SL1 have little effect on viral growth.

To clarify this point we added the following: "The apical loop shows very poor conservation, and, indeed, is exposed to the solvent, with no specific contacts with Ago2 (Fig. 1F and 1G)."

2. While PCBP2 has been reported to affect a switch from translation to replication, the effects were modest in full-length viral RNAs (ref. 15). Other RNA binding proteins have been implicated in the switch from translation to replication. In the absence of showing direct effects of PCBP2 on the switch, the authors may tune down or expand their discussion on PCBP2 being important for the switch from translation to replication.

We appreciate this comment. We agree that our experiments with isolated components do not provide functional data for the translation/replication switch. miR-122 Site-2 is immediately adjacent to Stem loop II, which is a fundamental part of the IRES that latches onto the 40S ribosomal subunit. We thus hypothesize that mutually exclusive binding of Ago2:miR-122 and PCBP2 at this site is well positioned to modulate translation, but it is certainly possible that other RBPs could play similar or even more prominent roles.

The manuscript was adjusted to better indicate PCBP2 is only one proposed contributor to the switch between translation and replication. Specifically, we changed the text to read:

“For HCV, one proposed mechanism to alternate translation and replication involves Ago2:miR-122 competing for binding at the viral 5' end with another host-protein, PCBP2”

and,

“Our data raise the possibility that the switch between replication and translation involves occupancy of either Ago2 or PCBP2, or other RBPs, at Site-2.”

and,

“Further studies will be required to determine how Ago2:miR-122, PCBP2, and other host factors interact with the HCV IRES and host translational machinery”

Reviewer #2 (Remarks to the Author):

The Ian MacRae group report on the structure of the 5'-end of HCV RNA bound to the Ago2:miR-122 complex and undertake structure-guided functional studies to deduce mechanistic insights into viral-host interactions that contribute to viral fitness. The structure outlines the binding of the HCV 5'-end containing stem-loop 1 to Ago2:miR-122 at site-1, thereby masking the 5'-end of viral RNA from enzymatic attack by nucleases and phosphatases. The authors identify the pairing between miRNA and HCV RNA, as well as viral RNA-Ago protein contacts, establishing several novel intermolecular interactions. The authors also identify a previously unidentified nucleotide binding pocket that locks Ago2 in a widened conformation and curbs lateral diffusion of Ago2 on target RNAs. They further show that a second Ago2:miR-122 can bind to adjacent site-2 on HCV RNA and protein PCBP2 competes for this site, suggestive of a molecular switch governing partitioning between regulation and translation.

This is a very important paper on how a viral RNA hijacks miRNA targets, with broad implications for the field. Both the structure and follow up functional experiments together with the writeup meet the highest standards. I strongly recommend publication of the paper in its present form after the authors address two minor comments listed below.

Thank you for the appreciative and encouraging comments.

Minor comments:

Did the authors attempt to crystallize Ago2:miR122 bound simultaneously to sites-1 and-2 on the 5'-end of HCV RNA, or alternately Ago2:miR122 bound to site-1 and PCBP2 bound to site-2 on the 5'-end of HCV RNA. It may be even possible to include the IRES RNA in the HCV RNA construct and solve the structures by cryo-EM. A comment regarding the feasibility of such an effort should be included in the Discussion.

We indeed did attempt to crystallize the Site-1 Site-2 complex, but these efforts were unfruitful. Recently, however, we have used cryo-EM to visualize the complex of two Ago2:miR-122 miRNPs bound to the full 5' UTR and the 40S ribosomal subunit. This study is still ongoing, but we are optimistic the work will illuminate how Ago2 interacts with both the HCV IRES and host translational machinery.

The Discussion section now explicitly reads: "Further studies will be required to determine how Ago2:miR-122, PCBP2, and other host factors interact with the HCV IRES and host translational machinery."

The later panels appear to be mislabeled in Figure 4 and should be corrected.

Thank you for the comment. We corrected the figure.

Reviewer #3 (Remarks to the Author):

Gebert et al.

A viral RNA structure that hijacks microRNAs

In this manuscript, the authors have further characterized the molecular details of the miR-122 interaction with the 5' end of the HCV genomic RNA. The 5' end of the HCV RNA contains two miR-122 binding sites upstream of an extended secondary structure serving as IRES. Interestingly, site 1 contains a short stem-loop that is not complementary to miR-122 and loops out. The interaction is stabilized by complementarity 3' of the short stem-loop. Gebert et al. have used biochemical approaches and report that deletion of the short stem-loop, as well as the individual miR-122 target site destabilize the HCV RNA. The miR-122-Ago complex at the very 5' end of the HCV RNA protects it from XRN1 exonucleolytic degradation. To unravel the underlying mechanisms, the authors solved the crystal structure of a ternary Ago2-miR-122-HCV (1-29) structure. From the structural data, they conclude that Ago2-miR-122 masks the 5' end and protects it from XRN1-mediated degradation. Furthermore, the short stem-loop holds Ago2 in an open conformation, which is supported by a specific nucleotide on the HCV target site (t9). Interestingly, these motifs prevent Ago2 from lateral diffusion, which is known from 'regular' miRNA target site recognition on host mRNAs. The second miR-122 binding site immediately downstream of the first target site further restricts lateral diffusion and the 5' end is efficiently protected. Finally, the authors report that the translational factor PCBP2 and the Ago2-miR-122 complex can bind simultaneously to the viral RNA allowing for target RNA protection and translation at the same time.

This is a well-written and very clear manuscript. It reports novel biochemical findings in an area that has been studied extensively during the past years. The reported crystal structure provides a clear and convincing model how and why the Ago2-miR-122 complex associates with the 5' end of the HCV RNA. Particularly, the model of restricting lateral diffusion appears appealing. I have only a few minor points that should be considered.

Thank you for these encouraging comments.

1. The authors propose a novel pocket in Ago2 that accommodates the unpaired target nucleotide t9/G21. Is this really a specific pocket or simply space where the flipped-out nucleotide is positioned?

We believe the t9 pocket is functional because experiments with an abasic nucleotide in the 21 position show effects on both target dissociation (Fig. 3E, 4E) and protection from XRN1

(Fig. 4B), indicating interactions with the pocket modulate Ago2 dynamics. We suspect the t9 pocket is specific for purines, especially G, because electron density unambiguously shows the t9/G21 purine base favorably stacked between P601 and R814 with the N2 amine (specific to guanine) near hydrogen bonding distance of two carbonyls in the protein main chain (Fig. 3C,D). Additionally, G at position 21 is well-conserved in HCV isolates, with A the second most observed nucleotide.

This structure seems to be unique to the HCV target RNA and thus might not be relevant for all other host miR-target interactions. Thus, a positive selection for such a pocket is unlikely. Are there other such target sites known on the human transcriptome?

The t9G/SL1 structure may indeed be unique to HCV. Early studies indicated A (but not G) at position t9 may be favorable for miRNA-target recognition (PMID: 15652477), but to our knowledge, host miRNA target sites with a t9G and tandem stem-loop resembling SL1 have not been reported. Additionally, use of the t9 pocket has not been observed in any previous Argonaute:guide:target crystal structure.

On the other hand, residues composing the t9 pocket are almost perfectly conserved between diverse animal species (Fig. S5B). It is therefore possible that t9G/SL1 structures exist and function in some animal transcriptomes. In this scenario, HCV did not necessarily invent the t9G/SL1 motif, but instead may have borrowed it from a class of endogenous miRNA target sites that has yet to be recognized. We note that such sites may be prone to being overlooked by traditional target site searches because secondary structure is generally considered a deterrent to Ago2:miR:target interactions (PMID: 26267216), and yet if secondary structure is not considered the seed- and supplementary-matching regions of the target site would appear distant from each other in the mRNA primary sequence. Indeed, considering the apical loop of SL1 is fully solvent exposed, seed and supplementary target regions could be separated by extended loops or other intervening structures in the place of the minimal loop in SL1 of HCV. Thus, our results indicate structures resembling t9G/SL1 could allow Ago2 to bridge regions distant from each other in the primary sequence of structured RNAs. It is our hope that our results may bring some attention to the possibilities.

2. The title is too broad. It is known that the HCV 5' end interacts and hijacks miR-122. This is not new. Furthermore, it only hijacks miR-122 in liver cells. The title should therefore be more specific.

We appreciate this suggestion and changed the title to: "A structured RNA motif locks Argonaute2-miR122 onto the 5' end of the HCV genome"

REVIEWERS' COMMENTS

Reviewer #1 (Remarks to the Author):

The authors have carefully addressed my minor questions. This study is a masterpiece in the field of microRNA-mediated gene regulation.

Reviewer #3 (Remarks to the Author):

In the revised version of their manuscript, Gebert et al. have adequately addressed the few minor points that I had raised on their previous version.